Agricultural big data and methods and models for food security analysis—a mini-review

Ammar Khalil A. 1
Kheir Ahmed M.S. drahmedkheir2015@gmail.com a.kheir@biosaline.org.ae 1 2
Manikas Ioannis 3
1 International Center for Biosaline Agriculture, ICBA , Dubai , United Arab Emirates
2 Soils, Water and Environment Research Institute, Agricultural Research Center , Giza , Egypt
3 Faculty of Business, University of Wollongong in Dubai , Dubai , UAE , United Arab Emirates
Adhikari Kabindra
Electronic publication date: 2022 Jun 29
Publication date: 2022
Volume: 10
Electronic Location ID: e13674
Received 2022 Feb 21; Accepted 2022 Jun 13
Copyright: ©2022 Ammar et al.
Copyright year: 2022
Copyright holder: Ammar et al.
License: This is an open access article distributed under the terms of the Creative Commons Attribution License, which permits unrestricted use, distribution, reproduction and adaptation in any medium and for any purpose provided that it is properly attributed. For attribution, the original author(s), title, publication source (PeerJ) and either DOI or URL of the article must be cited.
License URL: https://creativecommons.org/licenses/by/4.0/

Keywords: United Arab Emirates, Data extraction, Data infrastructure, Gaps, Challenges, Multi-model approach, Analysis, Visualization

Funding: Ministry of Education of the United Arab Emirates through the Collaborative Research Program Grant 2019, under the Resilient Agrifood Dynamism through evidence-based policies (READY) project 1733833 This study was funded by the Ministry of Education of the United Arab Emirates through the Collaborative Research Program Grant 2019, under the Resilient Agrifood Dynamism through evidence-based policies (READY) project (Grant number: 1733833). The funders had no role in study design, data collection and analysis, decision to publish, or preparation of the manuscript.

==============================
Background

Big data and data analysis methods and models are important tools in food security (FS) studies for gap analysis and preparation of appropriate analytical frameworks. These innovations necessitate the development of novel methods for collecting, storing, processing, and extracting data.

Methodology

The primary goal of this study was to conduct a critical review of agricultural big data and methods and models used for FS studies published in peer-reviewed journals since 2010. Approximately 130 articles were selected for full content review after the pre-screening process.

Results

There are different sources of data collection, including but not limited to online databases, the internet, omics, Internet of Things, social media, survey rounds, remote sensing, and the Food and Agriculture Organization Corporate Statistical Database. The collected data require analysis (i.e., mining, neural networks, Bayesian networks, and other ML algorithms) before data visualization using Python, R, Circos, Gephi, Tableau, or Cytoscape. Approximately 122 models, all of which were used in FS studies worldwide, were selected from 130 articles. However, most of these models addressed only one or two dimensions of FS (i.e., availability and access) and ignored the other dimensions (i.e., stability and utilization), creating a gap in the global context.

Conclusions

There are certain FS gaps both worldwide and in the United Arab Emirates that need to be addressed by scientists and policymakers. Following the identification of the drivers, policies, and indicators, the findings of this review could be used to develop an appropriate analytical framework for FS and nutrition.

Introduction

Global hunger due to climate change (Schmidhuber & Tubiello Francesco, 2007), pandemics (Kuehn, 2020), rapid population growth, dietary changes, as well as limited natural resources (Dutilleul, 2012; Janssens et al., 2020) has increased, which will make accessing nutritious and affordable food difficult in the future (Janssens et al., 2020). Cleaner production is essential for sustaining this expanding need for food production, but all the natural resources are under threat (Wunderlich & Martinez, 2018). Big data is required to investigate food and nutrition security due to population growth, global hunger, and widespread food demand (Jin et al., 2020). Big data is a term that can be defined in various ways but always refers to a large amount of data. Data is frequently produced in large quantities from various sources, necessitating the development of new tools and methods, such as powerful processors, algorithms, and software, to handle it (Marvin et al., 2017). Big data applications can be found throughout the food supply chain (FSC), from farm to fork, and can maximize production and ensure all food security (FS) dimensions and measurements are included. Online databases, omics profiling, the internet, sensors (Bouzembrak et al., 2019), mobile phones, social media (SM), video monitoring (Subudhi, Rout & Ghosh, 2019), portable devices, and sensors using Internet of Things (IoT) tools (Pal & Kant, 2019), geographic information system (GIS), remote sensing (RS) (Strawn et al.), survey (Delphi rounds), and blockchain method (George et al., 2019; Shaikh et al., 2019) are just some examples of the global data sources. The source type varies by region based on availability, type of required data, relative experts, and scientific background. Furthermore, rapid population growth and rising food demand necessitate the use of quick, digital, and reliable data sources to ensure all FS dimensions and measurements are included (Fritz et al., 2019). However, these sources still require global attention to promote and enable their use in various environments and cultures. Consequently, stakeholders have identified five key challenges impeding the effectiveness of agri-food system collaborations (OECD, 2021): (1) a lack of political visibility and prioritization, (2) a lack of long-term investment in statistics and data, (3) challenges in political economy, (4) limited skills and experience in using such technologies, and (5) access gaps to new data sources.

Apart from this, the collected dataset should consider all FS dimensions and measurements (i.e., availability, access, stability, and utilization) (Jones et al., 2013) (Fig. S1). The data should include, but not be limited to, agricultural production, food loss and waste, food supply sufficiency, agricultural infrastructure, population growth and Democratic Domestic Product (DDP), agricultural food costs, household income, water availability and quality, soil properties and biodiversity, human health and diet, consumer behavior, climate change scenarios, demographic changes and stress, market access, imports, and common crops in the specific area. The data should cover FS based on the food system analytical framework of the Food and Agriculture Organization (FAO), which is dependent on food and nutrition security (FNS) economic, environmental, and social factors (Food and Agriculture Organization, 2018). However, most studies on FS have focused solely on availability and affordability, resulting in gaps in data collection and limitations in FS quantification (Nkunzimana et al., 2018). Various countries, including the Gulf Cooperation Council (GCC) and poor and low-income countries, suffer from a lack of agricultural and FS statistics, even though sound decisions are based on accurate data and information. Despite their efforts, such countries continue to face several limitations, including a lack of household and farm survey data, large and long-term data, and data analysis and processing (Food and Agriculture Organization, 2021). Current data collection is primarily focused on national sources with varying degrees of coverage and accuracy, using surveys and operational records such as trade data. That information is frequently disseminated as statistical output, with little or no interpretation or analysis. As previously stated, one significant gap in food and nutrition security data is the lack of indicators relating to the quantities of various foods consumed to determine the adequacy of nutrient intake at both household and individual levels.

Understanding the long-term drivers of FS and how they interact is necessary for policymakers to make informed decisions about today’s policies for tomorrow’s FS (Van Meijl et al., 2020). Model-based scenario analysis is widely regarded as the appropriate tool given the complexity and uncertainty of multi-dimensional FS (Godfray & Robinson, 2015). This article stated that more than 91 household models classified as statistical, optimization, Computational General Equilibrium (CGE), simulation integrated, and simulation biophysical models were used for FS and considered only the first dimension (availability) and did not cover the other dimensions (i.e., affordability, stability and utilization) (Nicholson et al., 2021a). Such lack of inclusion is primarily due to various factors, including dataset availability, the power and type of model used (dynamic, statistical, etc.), and the purpose of the study. A multi-model ensemble can solve this problem and may capture all FS dimensions (Kheir et al., 2021a; Kheir et al., 2021b; Martre et al., 2015), but this approach has received less attention thus far, resulting in a global gap in the use of modeling to address FS issues. System dynamics (SD) refers to a scientific framework for dealing with complex, nonlinear feedback systems. The book entitled ‘Limits to Growth’, published in 1972 (Meadows et al., 1972) modeled for the first time the long-term risk of FS that would arise from the complex relationship between capital and population growth within the planet’s limits, using the World3 System Dynamics model (Nicholson et al., 2021b). Furthermore, machine learning (ML) models can work well with large datasets and have many advantages not found in other models (Abiodun et al., 2018; D’Amore et al., 2022), but they have received little attention thus far and require much attention in global FS studies (Chamara et al., 2020). The ML techniques can be used to automatically collect data using statistical or computational models, which can aid in accurately identifying factors and improving performance (Okori & Obua, 2011). In various languages, ML has a significant impact on sentiment analysis and text classification (Marie-Sainte et al., 2019). Opinion mining and sentiment analysis are techniques for analyzing people’s opinions, evaluations, sentiments, attitudes, and emotions from textual datasets (Onan, 2020c). There are numerous methods for text classification, opinion mining, and sentiment evaluation available in the literature (Onan, 2020a; Onan, 2020b; Onan, Korukoğlu & Bulut, 2016; Onan & Toçoğlu, 2021). However, the most widely used text classification techniques are lexicon-based, ML-based, and rule-based methods (Onan, 2021), with deep learning approaches not being used for feature selection or sentiment analysis, which necessitates much attention in FS studies. Consequently, while the integration of statistic, dynamic, ML, and deep learning models is very important in big data assessment and global FS studies, it has received less attention thus far, creating a gap that needs to be filled.

Regarding the position in the United Arab Emirates (UAE), unfortunately, there is a significant gap in large data sources, collection, and analysis. Furthermore, long-term investment in data statistics, digital skills, and sufficient data in science, technology, engineering, and mathematics (STEM) has been lacking. In addition, UAE lacks FS access, stability, and utilization of big data and methods and models based on FS dimensions. Therefore, a review study is required to identify the detailed background of big data and data analysis methods and models for FS on a global and regional scale to quantify the related gap and provide policy recommendations to fill it. Thus, this review screens the global and local big data and methods and models for FS analysis, highlights related challenges in the UAE, and suggests potential solutions.

Survey methodology

Many authors have emphasized the importance of conducting a literature review because they consider it a valuable and qualified source of information that summarizes and adds to the body of knowledge in a particular field of study (Denyer & Tranfield, 2009; Knopf, 2006; Tranfield, Denyer & Smart, 2003). The best literature review should reveal, assess, and structure the relevant literature on the intended topic, as well as combine it with a critical analysis of various arguments in the literature (Denyer, Tranfield & Van Aken, 2008; Tranfield, Denyer & Smart, 2003). The goal of this study was to conduct a literature review to identify the global big data, methods, and models for FS and investigate how they can be used to improve FS levels globally, as well as in UAE (Denyer, Tranfield & Van Aken, 2008; Tranfield, Denyer & Smart, 2003). The least level of bias was ensured via a comprehensive literature inspection of the available published studies to provide an audit pathway from the decisions of the reviewers to the actions and conclusions (Munn et al., 2018; Tranfield, Denyer & Smart, 2003). Furthermore, selecting the research methodology required identifying, analyzing, and synthesizing the selected secondary data sources related to FS big data, methods, and models across a wide range of contexts and disciplines to provide a comprehensive understanding based on the fit to the review’s specified questions. According to (Denyer & Tranfield, 2009; Munn et al., 2018; Tranfield, Denyer & Smart, 2003), producing good and comprehensive systematic reviews is crucial for driving research, developing new research baselines, and opening multiple pathways for future research. As a result, a systematic literature review research method was selected to achieve the research objectives. Based on the approaches described by Denyer & Tranfield (2009), Munn et al. (2018), Tranfield, Denyer & Smart (2003), we conducted the review through five steps to ensure replicability and transparency, as detailed in Fig. 1. The research began with the formulation of a research questions with specific characteristics, such as being purposeful and specific. The scope and focus of the review were then defined. The goal of this study was to conduct a systematic literature review to identify the big data, methods, and models of FS in the global and UAE contexts. To answer the main research questions, this article provides a critical review of the existing literature published in Scopus and Web of Science databases (Martín-Martín et al., 2018). The following topics have been explored in this review of literature: (1) data extraction tools, (2) data format and infrastructure, (3) potential and limitations of agricultural big data (AgBD), and (4) FS methods and models. Thus, this article provides a reference for policymakers and practitioners, as well as a roadmap for future research, by highlighting the concerns in the areas mentioned above.

Figure 1 Methodology and protocol of the systematic literature review.

The second step involved creating a specific research criterion to ensure that the research sources chosen were sufficient and comprehensive enough to capture all the major points that adequately answer the research questions (Denyer & Tranfield, 2009). The necessity of understanding big data for FS in both the global and UAE contexts, with a strong emphasis on avoiding any source of bias during the selection process, was the key research gap that drove this study. As a result, the databases Scopus and Web of Science were used (Martín-Martín et al., 2018). Big data, methods, models, FS availability, FS access, FS stability, FS stability, and food infrastructure were among the keywords used. The keywords were chosen after a thorough examination of the most relevant concepts in the literature that affect each of the four FS dimensions. In July 2021, the research sources were chosen, and the title, abstract, and full-text searches for keywords were enabled. To find the available literature, several keywords were identified (Cooper et al., 2018). Primary and secondary keywords were used in the search strings. The purpose of using multiple strings was to cover as many articles as possible that dealt with the topic of FS or any of its four dimensions. The review was then subjected to specific exclusion and inclusion criteria to produce high-quality evidence (Tranfield, Denyer & Smart, 2003). To ensure that the review has a high quality, a reasonable number of articles were selected for in-depth analysis based on a set of exclusion and inclusion criteria (Fig. 1). Within the time frame (2010–2021), only peer-reviewed journal articles written in English were included in the review. Strict selection criteria were applied to the first search pool to maintain research transparency and ensure the selection of relevant material that answers the research questions (Kelly, Sadeghieh & Adeli, 2014; Xiao & Watson, 2017). After removing duplicated articles from both databases, a total of 130 articles were chosen for the review.

The fourth step entailed analyzing the selected 130 articles individually, summarizing and listing all big data, methods, and models for FS analysis, then synthesizing the extracted information from all sources to create new knowledge (framework), listing the similarities between all resources, and extracting the major insights globally and within the UAE context (Denyer & Tranfield, 2009; Tranfield, Denyer & Smart, 2003). The models and methods for FS analysis in each of the 130 articles were summarized using Microsoft Excel. The aggregative approach was then used for synthesis. The findings section includes a detailed report of answers to the following research questions: (1) data extraction tools, (2) data format and infrastructure, (3) potential and limitations of AgBD, and (4) FS methods and models. After that, the synthesis process was used to create a comprehensive framework that models big data and FS methods and models.

Data extraction tools

Data extraction tools (global context)

The literature review showed different sources of data extraction, including online databases, smartphones, the internet, sensors, omics, social media (SM), Internet of Things (IoT), geographic information system (GIS), satellite images, web mining, the Food and Agriculture Organization Corporate Statistical Database (FAOSTAT), governmental dataset, statistical yearbooks as well as blockchain technology (Bouzembrak et al., 2019; George et al., 2019; Marvin et al., 2017; Pal & Kant, 2019; Strawn et al.; Subudhi, Rout & Ghosh, 2019). Online databases used widely in food safety (Jin et al., 2021; Marvin et al., 2017) covered only one dimension of FS (utilization) and neglected the other dimensions.

Smartphones are widely used in agriculture due to their ability to collect data, ease of mobility, which corresponds to the nature of farming, and low cost (Mendes et al., 2020; Pongnumkul, Chaovalit & Surasvadi, 2015; Thar et al., 2021). Nowadays, more than two billion people worldwide use smartphones, and this number is rapidly increasing, allowing the use of smartphones as important data sources in agriculture and FS (eMarketer, 2014). The numerous built-in sensors are among the factors that improve the smartphone’s ability to assist users with various tasks. Cheap smartphones may be a viable option for farmers who lack access to current agricultural information (e.g., market, weather, and crop disease news) and assistance from agricultural experts and government extension workers (Wolfert et al., 2017). Smartphones have recently been used in agriculture for various purposes, including food safety (Alfian, Syafrudin & Rhee, 2017; Shan et al., 2020; Ye et al., 2020), protein content determination (Silva & Rocha, 2020), food contaminant detection (Liu et al., 2017), weather and climate change reporting (Caine et al., 2015), as well as for agricultural and rural development (Donovan, 2017). Smartphones are the most important tools for receiving and recording terminal data (Guo et al., 2019). However, based on the literature, we observed that very little attention had been paid to understanding the various types of information communicated via smartphones, how farmers access this information, and the possible factors influencing the use of smartphones. Furthermore, smartphone applications did not cover all dimensions of FS (i.e., availability, access, utilization, and stability), necessitating a great deal of attention on the global, national, and individual levels.

SM sites are websites that allow users to create profiles (Hether, Murphy & Valente, 2014), share content, and engage in discussions to facilitate communication and community engagement (Bertrand et al., 2021). SM has been widely used to collect food safety data (Wang et al., 2016). Making intelligent decisions based on social big data refers to the techniques, technologies, systems, and platforms that help organizations better understand their data and make better decisions (Wang et al., 2016). FS-related discussions, opinions, and online questionnaires can be collected using SM platforms such as Facebook, Twitter, and YouTube (Soon, 2020). Web mining is a popular method for collecting and analyzing SM data. By analyzing customer sentiments and opinions, SM data could be used to improve client behaviors, raise public awareness, and understand public perceptions of FS (Yuan et al., 2020).

RS data can be used in agriculture for monitoring crop growth, development, and harvesting, and improving the existing monitoring systems, all of which contribute to improved agricultural product quality (Friedl, 2018; Singh et al., 2020). Data of RS images in the European Union (EU) could be accessed by Sentinel-2 satellites for various applications in agriculture and FS (Kussul et al., 2020; Phiri et al., 2020). The FAO GeoNetwork and RS database include grids and layers for classifying soil, water, and climate for monitoring food safety and security (Jin et al., 2020). Hattenrath-Lehmann et al. (2018) used RS as an early warning system for shellfish safety, while the US Department of Agriculture used them to detect food contamination (https://cris.nifa.usda.gov/). However, using the RS approach in big data for other FS dimensions such as availability, access, and stability has received less attention thus far, necessitating a great deal of attention on a large scale from stakeholders.

IoT is the interconnection of devices, sensors, machines, and computing devices through internet mediums (e.g., Wi-Fi, Bluetooth and Radio Frequency Identification (RFID)). This technology has the potential to make the food chain more efficient, safer, and sustainable in the near future. Kaur (2021) modeled sustainable FS based on IoT technology and determined how to design a long-term FS system in India, where the government ensures FS for all through a public distribution system (PDS). The study also made a novel attempt to incorporate IoT into the design of the PDS to ensure FS, with IoT factors being modeled using Total Interpretive Structural Modeling (Fuzzy-TISM). FS can be ensured using IoT as it provides traceability, transparency and accountability, decreasing food waste and ensuring food quality from harvest to consumption (Nukala et al., 2015). For more accurate results, IoT could be combined with technological enablers such as artificial intelligence, robotics, blockchain, and RFID. The use of these technologies will help reduce food waste and enable better planning of distribution networks, lowering the overall supply chain carbon emissions (Irani & Sharif, 2016). Various studies have investigated the importance of IoT in FS, but they have focused only on some dimensions, implying that all FS dimensions require much attention. Ding et al. (2014) proposed a conceptual model to investigate the interdependencies between different functions and information shared in FSC. Fan et al. (2015) proposed a big data analytics-based algorithm to improve crop yield prediction accuracy. Masiero (2015) emphasized the importance of digitizing the food distribution function and the role of e-governance in preventing food fraud in Kerala. The role of information sharing in fresh FSC was examined by Nakandala et al. (2017), who identified the information needs of various supply chain entities. Table 1 summarizes various enabling factors for an IOT-driven sustainable FS system culled from the literature. However, understanding the relationships and their effect among different technologies is critical for designing an IoT-driven FS system that is also sustainable. The FS system is a multi-level, multi-stakeholder problem. The integration of various enabling factors is difficult to establish. Furthermore, the impact of one relationship may differ from that of another. As a result, the magnitude of the impact is also a crucial factor to consider. From the standpoint of policymaking and implementation, such factors must be structured in a conceptual model to ensure long-term FS. Moreover, such technologies with attributed enablers should cover all dimensions of FS, rather than just one; thus, a gap needs to be filled from both the global and stakeholder perspectives.

Table 1 Enabling factors of IoT-driven sustainable food security (modified after (Kaur, 2021)).

Parameter	Reference	Role in sustainable food security	
Yield prediction based big data	Engen et al. (2021), Fan et al. (2015) and Kamath et al. (2021)	Assist in the procurement process and the distribution of food resources across different regions.	
Delphi survey	Allen et al. (2019), Markou et al. (2020) and Vogel et al. (2019)	It can aid in the procurement and distribution of goods in a decentralized and distributed manner.	
Traceability based Blockchain	Lin et al. (2017), Xiong et al. (2020) and Yadav et al. (2021)	Avoid food losses, shrinkages, and fraud in FSS	
Mobile application for crop details	Ahmed & Reddy (2021), Meeradevi & Salpekar (2019) and Yang & Xu (2021)	Crop yield, diseases prediction, horticulture research and policy designing	
Robotics technology	Asseng & Asche (2019) and Asseng et al. (2020)	Food production and quality without farmers	
Sensors and image processing	Eerens et al. (2014), Lew et al. (2020) and Rasti et al. (2021)	Ensure better quality control, and higher yield	
Sharing information-based channels	Singla, Nishu & Deepika (2020) and Wolfert et al. (2017)	Better supply chain coordination is aided by information sharing. It also helps supply chain partners build trust.	
Refrigeration IoT interface	Talavera et al. (2017)	The temperature can be adjusted depending on the type and quantity of stock in the refrigerator.	
Food AI package before date	Mavani et al. (2021)	Decreasing food waste and ensuring food safety	
Policy improvement using technology	Jeevanandam et al. (2022) and Masiero (2015)	FSS monitoring and quality control	
e-farm marketing	Masiero (2015)	Avoid losses, maintain food and exclude the intermediate retailers	
Consumption pattern simulations	Christensen et al. (2018) and Yang et al. (2020)	Assist policy-makers in designing a FS system that is appropriate for population consumption behavior. Modeling the pattern of power consumption using a single sensor	
Encoded digital data	Golan, Jernegan & Linkov (2020)	Tracking the goods movement throughout the supply chain.	
Cloud computing optimization (Google Collaboratory, Azur, IBM, AWS)	Castelvecchi (2017), Christensen et al. (2018), Langmead & Nellore (2018), Satyanarayanan (2019) and Vanderroost et al. (2017)	Saving time, reducing food losses, and keeping high quality	

Data extraction tools (The UAE context)

Despite the lack of data for measuring FS at the household or individual levels, several data sources are available to analyze and monitor the UAE’s FS progress at the national level. For example, data on most of the Suite of FS Index indicators for the UAE are available at FAOSTAT (http://www.fao.org/faostat/en/#data/FS). These statistics are mostly available as three-year averages. The Economist Intelligence Unit (EIU) publishes the Global Food Security Index (GFSI) score for the UAE every year as part of its multi-country FS monitoring. The available data from EIU and FAO on the FS dimension indicators used to construct the GFSI and Suite of FS Index can be used as inputs for deriving other simple and multi-dimensional measures of FS.

Data on the UAE’s Food Balance Sheet can also be obtained from the FAOSTAT (http://www.fao.org/faostat/en/#data/FBS). The dataset contains information, among others, on food supply (kcal/capita/day), protein supply quantity (g/capita/day), domestic production, import and export, feed and other non-food uses, food stock variation, tourist consumption, and food losses. Furthermore, data on food price indices and food inflation can also be retrieved from the FAOSTAT (http://www.fao.org/faostat/en/#data/CP).

Similar datasets at disaggregated levels can also be obtained from national offices (e.g., for Dubai, it can be obtained from the Dubai Municipality). The various organizations within the UAE, such as the Ministry of Food Security, Ministry of Health and Prevention, Dubai Municipality, Abu Dhabi Agriculture and Food Safety Authority, and the Federal Competitiveness and Statistics Centre, could be consulted for obtaining a variety of data that can be used as inputs for estimating some of the FS indicators.

Delphi survey rounds could be considered an effective data source (Allen et al., 2019; Markou et al., 2020) for collecting the required data to validate the analytical framework and quantify sustainable FS in the UAE.

Data format and infrastructure

FS data can be unstructured or structured and stored in various formats, including TXT, JSON, and CSV. For instance, Singh, Shukla & Mishra (2018) collected SM data from Twitter in JSON and TXT formats, and then implemented the parsing method to covert JASON data to CSV data. There are also various formats for big data, such as raster and vector formats (SHP, TIF, CN, and NetcDF) (Ghiringhelli et al., 2017; Limbachiya & Gupta, 2015). On the other hand, Song et al. (2020) stored the data in relational databases with different attributes as a list of rows. Alfian, Syafrudin & Rhee (2017) used NoSQL and SQL databases to store IoT-generated sensor data with a large unstructured format and continuous data-generation characteristics. They also developed a real-time food quality monitoring system that employs sensor data from a smartphone and stores it in the MongoDB database.

Supercomputing and cloud computing are two major components of data infrastructure (Yang et al., 2017b). To address the challenges associated with big data, supercomputing must be considered. The United States has long been committed to supercomputing to facilitate knowledge exchange between the Exascale Computing Project and the industrial user community (Witze, 2014). The development of supercomputing infrastructures is also a priority for the EU. So far, the EU has built eight supercomputing centers to enhance bioengineering applications (Butler, 1999). Tianjin, Jinan, Changsha, Shenzhen, Guangzhou, Wuxi, and Zhengzhou are China’s seven national supercomputing centers. The Chinese government has created a food safety traceability platform (Berti & Semprebon, 2018) that collects 31 provincial food traceability data and connects national supercomputing centers. Its goal is to achieve food traceability from farm to fork while also providing services to food producers, such as food traceability, security, and oversight.

To enable big data research, several cloud computing infrastructures need to be developed (Yang et al., 2017a). In 2019, the EU Food Nutrition Security Cloud project (https://cordis.europa.eu/project/id/863059) aimed to integrate European research infrastructure by bringing together FNS data to address diet, health, and consumer behavior, as well as sustainable agriculture and bioeconomy. The Guizhou Food and Drug Administration in China released the food safety cloud system in 2014. It has now been transformed into an intelligent food safety supervision system, an internet plus inspection system, a traceability certification system, and a big data platform for government enterprises, testing institutions, and other social age organizations (Tao et al., 2018). Despite the importance of supercomputing and cloud computing infrastructures as distinct big data environments, the Middle East and North Africa (MENA) and GCC regions remain uninterested, necessitating significant attention to assist them in addressing food insecurity.

Agricultural big data (potential, current status, and limitations)

To meet the demands of the rapidly growing population, which is expected to reach nine billion people by 2050 (World Population Prospects, 2015), agricultural production and FSCs must be optimized by producing and delivering efficient food, feed, fiber, and fuel (Abe, 2017; Asseng et al., 2018; Asseng et al., 2019). This goal has become more difficult to achieve due to urbanization, climate change (Ali et al., 2022; Ali et al., 2020; Ding et al., 2021; Kheir et al., 2019), and water scarcity (Kheir et al., 2021b). AgBD will be a key component of the second green revolution, which will be required to meet the demands of the growing population. Furthermore, the crop growth simulation modeling approach has been proven to be a useful tool for determining the impact of climate uncertainty on crop yields (Asseng et al., 2013; Ejaz et al., 2022; Shoaib et al., 2021). Many countries and commodity markets are already using AgBD to detect supply chain disruptions in commodity crops like wheat, rice, corn, and soybean (Bock & Kirkendall, 2017; George Hanuschak, 1993; Rosenzweig et al., 2013). Precision agriculture has progressed as a result of advancements in RS data collection, such as improved spatial and temporal resolution, spectral resolution, and a variety of sensor platforms (e.g., satellite, aerial, and ground-based) (Mulla, 2013). Precision agriculture recently demonstrated a significant increase in crop yield production (Loures et al., 2020; Singh et al., 2020). Spatial data mining techniques (e.g., hotspot detection) can be used with AgBD to identify crops produced in small geographic areas or a set of regions that are vulnerable to climate change and natural disasters (Jiang & Shekhar, 2017; Shekhar, Feiner & Aref, 2015; Vatsavai et al., 2012; Xie et al., 2017). Furthermore, consumer datasets and market manner can be used to improve food access and nutritional outcomes, and geo-social media can be used to detect and control food-borne illness outbreaks in real-time. AgBD could help agricultural decision-makers in four ways: descriptive, prescriptive, predictive, and proactive (Shekhar et al., 2017). The goal of the descriptive axis is to use AgBD data to characterize spatial and temporal variability in soil, land cover, crop, and weather characteristics, as well as to identify stressors, traits, and infectious disease risk factors that need to be better managed. The prescriptive way is to look for the required innovations for farm management. The predictive axis is a predictive analysis that uses historical datasets and integrated soil, crop, weather, and market models to forecast outcomes like crop yields and food insecurity. Predictive analytics can also be used to improve decision-making to forecast the spread of infectious agents and limit their impact on crops and livestock. Finally, the proactive axis includes crop development and stress observations from multiple farms across large regions and time scales.

The current state of AgBD can be divided into two categories: public and private data. Some examples of public AgBD are summarized in Table 2. Big data differs from one region to another based on various factors, including but not limited to the data availability, capacity building, and target of the study. Exploring sustainable FS necessitates detailed data covering all aspects of FS, putting pressure on decision-makers and scientists to initiate and prepare the necessary big data. To assist in filling this void, the big data paradigm should employ techniques, paradigms, and decision-making technologies, as illustrated in Fig. 2.

Table 2 Examples of public agricultural big data with related references.

Type	Source	References	
Meteorology and RS data	Cloud computing-based earth	AWS (2017)	
	Cloud computing-based Google earth engine	Google Earth Engine (2017)	
	Cloud computing-based NASA, NOAA	NASA (2017) and National Oceanic and Atmospheric Administration (2017)	
	Cloud computing-based labor statistics	United States Bureau of Labor Statistics (2017a)	
Survey	National Agricultural Statistics Services (NASS)	United States Department of Agriculture (2017b)	
Financial	National Water Economy Database (NWED)	Rushforth & Ruddell (2017)	
Scientific data	Scientific Research Centers in Agriculture	United States Department of Agriculture (2018b)	
Geospatial, water and soil	Natural Resources Conservation Service (NRCS)	United States Department of Agriculture (2018c)	
Sales and prices	Agricultural Marketing Services	United States Department of Agriculture (2018a)	
Marketing	World Agricultural Outlook Board	United States Department of Agriculture (2018d)	
Generic	Global Open Data for Agriculture and Nutrition	United States Department of Agriculture (2022)	
Notes.

Modified after U.S. Department of Agriculture (USDA).

Figure 2 Processing of big data paradigm.

Methods and Models used for FS analysis

FS has four dimensions: access, availability, stability, and utilization; thus, incorporating such dimensions into agricultural models necessitates careful consideration and deep efforts. There have been few studies that have linked agricultural models with FS dimensions and indicators in order to understand evolving intertemporal dynamics and assess the effects of agricultural system intensification (Laborte et al., 2009; Laborte, Van Ittersum & Van den Berg, 2007; Marín-González et al., 2018; Morris, 2003; Nicholson et al., 2021a). However, such studies have focused only on a few FS indicators, such as household outcome, and ignored other dimensions and indicators. Therefore, we reviewed more than 1,200 related articles on FS modeling at the household and regional levels to assess the frequency of use of various FS indicators and make future recommendations to close this gap. Optimization models were used in FS but only on a few indicators of food availability (Amede & Delve, 2008). Crop simulation models are used to predict crop yield as an indicator of food availability, either as regression models (Beyene & Engida, 2016; Bharwani et al., 2005) or as complex biophysical models (Lázár et al., 2015). For food consumption expenditures, other sophisticated models, such as Holden & Shiferaw (2004) and Louhichi & Gomezy Paloma (2014) were used. Scopus database was screened for approximately 1,250 articles related to household FS models, and 130 articles were reviewed for which FS indicators were summarized (Table S1). We also looked for studies that looked at the determinants of dietary diversity at the individual level (most commonly, among young children or women) or at the household level. Dietary diversity, or the number of different foods or food groups in one’s diet, has been linked to several measures of household socioeconomic status that are frequently used as indicators of food insecurity (Jones et al., 2013). As a result, dietary diversity is frequently used as a proxy and stand-alone indicator of household food insecurity. We searched Google Scholar for relevant studies that provide empirical evidence about the determinants of the Household Dietary Diversity Score (HDDS), Household Food Insecurity Access Scale (HFIAS), and Food Insecurity Experience Scale (FIES). For this purpose, we used the following items in search: diet diversity determinants (130 articles), household FS determinants (870 articles), and experience scale of food insecurity (250 articles). From the 1,250 articles, 130 articles were reviewed. Figure 3 shows the network and associations of different models used for FS. It was found that statistical models were the most prevalent, and all models covered only two FS dimensions: availability and access. The detailed descriptions of these models, including type, classifications, the related references, and calculations of FS dimensions are presented in Table S1. Even though, many models and methods are used to investigate global FS, there are insufficient related studies in the GCC, particularly in the UAE, requiring much attention using the best tools to achieve most FS dimensions.

Figure 3 Network visualization of the model number and types used in food security from literature over last 10 years.

Conclusions

Despite the importance of big data tools in FS, some challenges must be addressed first (Wang et al., 2016). The most common challenges for FSC-related data, according to most experts, are data quality, accessibility, findability, reusability, interoperability, and a lack of standardization. Farmers, for example, use a variety of farm management systems, making standardization of farm management data (such as variable names) a challenge. Because of the lack of standardized communication protocols, the data produced by IoT devices today can be difficult to interpret, communicate, and share, which may be one of the reasons for the limited adoption of IoT technology in food safety (Bouzembrak et al., 2019). Handling big data issues is difficult and time-consuming, requiring a large computational infrastructure to ensure timely data processing and analysis. Even though many organizations have adopted cloud computing as a solution, research on big data in FS using cloud computing technology is still in its infancy. Scalability, availability, data integrity, security, privacy, and legal issues are just a few of the research challenges that have yet to be fully addressed globally and in the UAE.

Statistical, optimization, CGE, simulation integrated, and simulation biophysical models and methods were used globally. However, it was discovered that such models only covered a subset of FS dimensions, namely availability and access, while recording limitations with other dimensions. In the future, this will necessitate the use of a multi-model approach to investigate FS because it will cover most FS dimensions while achieving higher accuracy. Thus, currently, there are some FS gaps in the global and UAE contexts that require significant attention from scientists and decision-makers. The global gaps could be summarized as follows: limited global dissemination of big data digital sources, lack of political visibility and prioritization, lack of long-term investment in data and statistics, lack of coordination and political economy challenges, limited access to new data sources, utilization and stability dimensions were not covered well, model complexity and uncertainty of multi-dimensional FS, limited studies on multi-model approach, and deep learning approach not being used. In the UAE context, in addition to the gaps mentioned in the global context, limited big data sources, lack of long-term investment in data and statistics, insufficient investment in agricultural research, insufficient studies in STEM, lack of modeling studies, and lack of ML and deep learning (DL) in data collection and analysis. Following the identification of the drivers, policies, and indicators, these findings could be used to develop an appropriate analytical framework for FS and nutrition.

Supplemental Information

Figure S1 Access, stability, availability, and utilization are the four main dimensions of food security

Data source: FAO. 2019. The State of Food and Agriculture 2019. Moving forward on food loss and waste reduction. Rome.

Licence: CC BY-NC-SA 3.0 IGO.

Click here for additional data file.

Table S1 Reviewed food security models based on FS dimensions

Click here for additional data file.

We would like to thank Editage for English language editing.

Additional Information and Declarations

Competing Interests

Author Contributions

Data Availability

The authors declare there are no competing interests.

Khalil A. Ammar conceived and designed the experiments, performed the experiments, analyzed the data, prepared figures and/or tables, authored or reviewed drafts of the article, and approved the final draft.

Ahmed M.S. Kheir conceived and designed the experiments, performed the experiments, analyzed the data, prepared figures and/or tables, authored or reviewed drafts of the article, and approved the final draft.

Ioannis Manikas analyzed the data, authored or reviewed drafts of the article, and approved the final draft.

The following information was supplied regarding data availability:

The raw data is available in the Supplemental File.

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
