# Peer review of "Agricultural big data and methods and models for food security analysis—a mini-review"

_PeerJ, doi:10.7717/peerj.13674_

## Round 0.1 · original submission · Major Revisions

I have completed the review of your article. It needs several major revisions. Please revise and resubmit.

Reviewer 1 ·

Basic reporting

The present study reviews Big Data, methods, and models in Agriculture for Food Security. I appreciate the work of the authors.

Specific Comments to the Authors:

1- After reading the manuscript, I see the title should be: Big Data, methods, and models in Agriculture for Food Security – A mini-review.

2- Keywords must be relevant for database search and different from those already appearing in the title. The function of keywords is to supplement the information given in the title. The title's keywords are automatically included in indexes, and keywords serve as additional pointers. The author should pay good attention to the keywords they provide.

3- From Lines 14-15, "about 150 articles were selected", and from Lines 331-332, "122 papers were reviewed", Line 131 "a total of 130 papers were chosen": which papers count is correct?.

4- Line 20: what do you mean about "Approximately 122 models were examined"?

5- Many abbreviations have been used without prior declaration through the manuscript, which made the revision uncomfortable, such as Line 75, 210, 274, and so on.

6- However, it is a review article; there is a lack of citations; see Lines 105-178.

7- Line 185: "the two research questions" please compare with Lines 128-131.

8- Lines 196-197: "Data of Landsat images could be accessed by Sentinel-2 in the European Union" this is wrong technically: how Landsat can be accessed by Sentinel-2?.

9- Line 269: (http://www.chinatrace.org/index.html) is not in English.

10- Many English corrections have to change (Big Data, big data, bigdata), and the majority of citations in text are incorrect; see Line 2016, 218.

11- By Section 2.2, it is not enough to add UAE to the title of the manuscript.

12- However, the manuscript reviews big data, which mainly refers to raster and vector data formats (shp, tif, cn, netcdf); they are not included in Section 2.3.

13- Generally, authors have to move from the literature-review style to the review-article style, for instance: https://doi.org/10.2136/vzj2015.09.0131

Experimental design

no comment

Validity of the findings

no comment

Additional comments

no comment

Reviewer 2 ·

Basic reporting

The review paper "Big Data, Methods, and Models in Agriculture for Food Security – A Review for Global and UAE Contexts" is very interesting because it investigates the available and required datasets, methods, and models for food security studies in arid environments. Given the importance of the current study as an initial and genuine start in food security studies, there is a significant gap in this type of research in both UAE and GCC countries. Furthermore, the paper identified food security gaps in the global and UAE contexts, which could serve as a foundation for future directions to close such gaps.

Therefore, I do recommend it for publication after addressing some comments below.


- Sometimes you mention big data and bigdata elsewhere, please be consistent through the text

- Lines 81-86 should include the most common ML models used for food security, as well as the relevant references.

- Headings number after the Introduction section looks confused, please check, and correct wherever. For instance, number 2 is the survey methodology, and this part should be separated from other headings.

- The section Agricultural Bigdata (potential, current status, and limitations), requires much information and references. There are many methods, models and datasets used in agriculture in different environments, please extend this section.

- Please correct any grammar errors and improve the language as needed.

Experimental design

1- The Survey Methodology is consistent with a comprehensive, unbiased coverage of the subject,

2- The sources are properly cited and quoted or paraphrased as needed.

3- The review organized logically into coherent paragraphs/subsections

Validity of the findings

1- There is a well-developed and well-supported argument that meets the objectives stated in the Introduction.

2- The conclusion identifies unanswered questions, gaps, and future directions.

Additional comments

The review paper "Big Data, Methods, and Models in Agriculture for Food Security – A Review for Global and UAE Contexts" is very interesting because it investigates the available and required datasets, methods, and models for food security studies in arid environments. Given the importance of the current study as an initial and genuine start in food security studies, there is a significant gap in this type of research in both UAE and GCC countries. Furthermore, the paper identified food security gaps in the global and UAE contexts, which could serve as a foundation for future directions to close such gaps.

Therefore, I do recommend it for publication after addressing some comments below.

- Sometimes you mention big data and bigdata elsewhere, please be consistent through the text
- Lines 81-86 should include the most common ML models used for food security, as well as the relevant references.
- Headings number after Introduction section looks confused, please check, and correct wherever. For instance, number 2 is the survey methodology, and this part should be separated from other headings.
- The section Agricultural Bigdata (potential, current status, and limitations), require much information and references. There are many methods, models and dataset used in agriculture in different environments, please extend this section.
- Please correct any grammar errors and improve the language as needed.

Reviewer 3 ·

Basic reporting

This is a very interesting paper; I recommend that it be published as a review of the broad and cross-disciplinary interest and within the scope of the journal, and the Introduction adequately introduces the subject and makes it clear who the audience is/what the motivation is. However, some comments below must be addressed first.
1- Abstract is OK, but at the end of line 24, you should add another sentence as a future direction message.
2- Lines 31-32, global hunger happened due to many factors not only population, income, and dietary changes change, but also due to climate change, pandemics, as well as limited natural resources. So, please add them with further references.
3- Line 34, Large dataset or Big dataset? And what is the difference? please be consistent thoroughly.
4- Line 53-54, please add a supplementary Figure for FS dimensions.
1- Lines 62-63 " Different countries, in particular poor and low income, suffering from limited data of agricultural, and food security statistics, even though sound decisions are based on accurate data and information," I think that you need to rewrite this sentence as GCC countries should be included in countries with limited data sources.
2- Lines 79-81, how can you prove that, add the appropriate reference. Furthermore, you need to extend this sentence with other sentences to explain the importance and significance of dynamic models in food security over different environments. There are many sources in the literature that explained that, so go through them and specify the knowledge gap.
3- Survey methodology, presented very well, thank you.
4- Line 263, add the appropriate reference
5- Section 2.4. Agricultural Bigdata (potential, current status, and limitations), to be consistent, please specify if it is for global or UAE context
6- Section 2.5. Methods and Models for FS Should add the word "used" so, the correct should be " 2.5.Methods and Models used for FS"
7- Line 377, add a future research need by mentioning the other subjects required for food security studies, such as analytical framework, drivers, indicators, and scenario development. This will assist scientists and policymakers inadequately addressing FS in the UAE.

Experimental design

1- The Survey Methodology is consistent with a thorough, unbiased examination of the topic.
2- The sources are properly cited and paraphrased or quoted as needed.
3- The review is divided into logical paragraphs and subsections.

Validity of the findings

1- The argument is well-developed and well-supported, and it meets the goals stated in the introduction, but further references are needed.
2- The conclusion identifies unanswered questions, gaps, but future directions are required.

Additional comments

Language needs improvement through the text.

---

## Round 0.2 · Minor Revisions

The manuscript would benefit from professional English proofreading. Phrases such as "Results showed that", etc. could be removed, and the writing could be more concise.

Reviewer 2 ·

Basic reporting

No comment

Experimental design

No comment

Validity of the findings

No comment

Additional comments

No comment

Reviewer 3 ·

Basic reporting

No comment

Experimental design

No comment

Validity of the findings

No comment

Additional comments

After responding to all of the comments, the paper improved significantly.

---

## Round 0.3 · Minor Revisions

Thanks for the corrections; however, the writing is still not good. I noted various grammar errors. Please revise and provide evidence of professional English proofreading.

---

## Round 0.4 · accepted · Accept

Thanks for the revisions.